# Hyperpolyglot LLMs: Cross-Lingual Interpretability in Token Embeddings

**Andrea W Wen-Yi**
Cornell University
andreawwenyi@infosci.cornell.edu

**David Mimno**
Cornell University
mimno@cornell.edu

## Abstract

Cross-lingual transfer learning is an important property of multilingual large language models (LLMs). But how do LLMs represent relationships between languages? Every language model has an input layer that maps tokens to vectors. This ubiquitous layer of language models is often overlooked. We find that similarities between these input embeddings are highly interpretable and that the geometry of these embeddings differs between model families. In one case (XLM-RoBERTa), embeddings encode language: tokens in different writing systems can be linearly separated with an average of 99.2% accuracy. Another family (mT5) represents cross-lingual semantic similarity: the 50 nearest neighbors for any token represent an average of 7.61 writing systems, and are frequently translations. This result is surprising given that there is no explicit parallel cross-lingual training corpora and no explicit incentive for translations in pre-training objectives. Our research opens the door for investigations in 1) The effect of pre-training and model architectures on representations of languages and 2) The applications of cross-lingual representations embedded in language models.

## 1 Introduction

Multilingual large language models (LLMs) have the potential to support transfer learning between languages with little to no additional training data (Lauscher et al., 2020; Wu and Dredze, 2019; Conneau et al., 2020; Winata et al., 2021). But we have limited theory for how LLMs represent meanings across languages. This work describes a mechanism for cross-lingual transfer learning by measuring the properties of the input embedding vectors. While most of the interest in LLMs rightly focuses on their ability to produce contextualized output, in this study we focus specifically on the lowest network layer: the initial token embedding layer. This layer often comprises a large percentage of the total parameters in a model. It also serves as the connection between human-readable strings and the latent vector representations that initialize the remaining layers. As such, the initial token embedding layer is both geometrically expressive and readily interpretable. We explore the initial token embeddings of two highly multilingual model families, XLM-RoBERTa (XLM-R) (Conneau et al., 2020) and mT5 (Xue et al., 2021). We find that mT5 discovers a universal, cross-lingual semantic space that assigns words (or word fragments) with similar meanings to nearby vector positions.[1]

Previous work has shown that algorithms exist to train multilingual word embeddings without explicit parallel text or glossaries (Ammar et al., 2016; Chen and Cardie, 2018). What is novel about the current work is the discovery that certain highly multilingual LLMs contain such an embedding *as an emergent property* without any explicit instruction to do so.

Explaining the factors that cause LLMs to find cross-lingual semantic embeddings would require pre-training experiments that are beyond the scope of this short paper. Describing these behaviors is, however, a necessary first step. At the same time, there is increasing attention on producing language technology for lower-resource languages (Kumar and Albuquerque, 2021), where the data-hungry methods that have been successful for high-resource languages may not be applicable. Creating predictive theories that explain the relationship between properties of pre-training data and representations learned by LLMs could lead us to build collections, architectures, and algorithms that most efficiently improve performance. This will be an extremely valuable step to enhance language technology for low-resource languages.

---

[1]Code is available at: https://github.com/andreawwenyi/hyperpolyglot

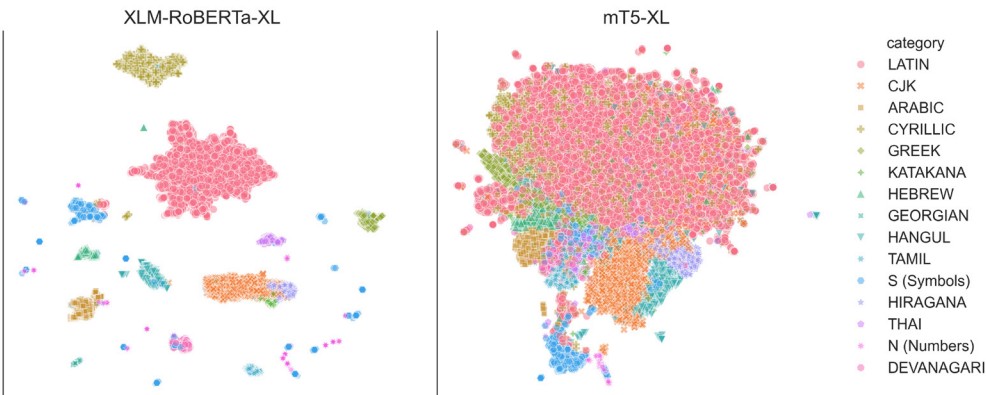

Figure 1: 2D Projections of input token embeddings. In XLM-RoBERTa-XL (left), writing systems are widely spaced; while for mT5-XL (right), vectors are more mixed.

## 2 Related work

Previous work has found interpretable patterns in feed-forward layers (Geva et al., 2021, 2022), self-attention (Mrini et al., 2020; Clark et al., 2019; Serrano and Smith, 2019), and input embeddings from the perspective of adversarial perturbation (Sato et al., 2018). In this work, we directly interpret the relative positions of the token embeddings through the lens of the vocabularies.

Prior to the explosion of contextualized language models, there was substantial interest in cross-lingual word embeddings (CLWE) (Ruder et al., 2019; Mikolov et al., 2013; Lazaridou et al., 2015). The goal is often to map monolingual word embeddings from different languages to the same shared space, so that words of the same meaning are close to each other. CLWE approaches involve some levels of supervised alignment (Faruqui and Dyer, 2014; Zou et al., 2013), seed dictionaries (Artetxe et al., 2017; Gouws and Søgaard, 2015) or adversarial training (Lample et al., 2018; Artetxe et al., 2018; Zhang et al., 2017; Miceli Barone, 2016). Contexualized embeddings from language models have also been used in combination with static word embeddings to improve alignments of cross-lingual word vectors (Aldarmaki and Diab, 2019; Zhang et al., 2021). Contrary to our findings for the token embeddings of LLMs, it was not clear that aligning word vectors is possible without some level of supervision, or to more than two languages at a time. Previous results also showed that CLWE approaches are sensitive to language pairs, where languages with large semantic or structural differences usually failed, such as English-Japanese and Spanish-Chinese (Xu et al., 2022).

## 3 Multilingual vocabularies

**Sub-word tokenization and writing systems** Initial embedding layer maps *tokens* to vectors. Most contemporary language models use sub-word tokenization schemes such as byte-pair encoding. These methods balance representational richness (providing distinct vectors for as many inputs as possible) with efficiency in limiting overall vocabulary size. While some methods produce tokens consisting of byte sequences that are not valid UTF-8 characters, in practice almost all tokens are valid, displayable sequences of Unicode characters, and a large number are recognizable words [2].

As it is difficult to assign tokens to specific languages (e.g. *war* can be an English noun or a German verb), we use Unicode metadata to define categories of tokens. For example, a is *LATIN SMALL LETTER A*. Most characters are letters (L), but vocabularies also include punctuation (P), numbers (N), and symbols (S, which include emoji). Letters are also marked by writing system, such as LATIN, CYRILLIC, or BENGALI. We define the *category* of a character as either its writing system (for letters) or its Unicode class for non-letters. A token can mix characters of different categories, but they typically have a most frequent category: the string doesn't contains letters and punctuation, but is primarily LATIN letters. We define the category of a token based on its majority character category. As a result, doesn't is classified as LATIN.

**Overlap between model vocabularies** While model families have distinct sets of vocabular-

---

[2]We find 39.93% of mT5 and 47.78% of XLM-R vocabularies in multilingual dictionaries provided by Lample et al. (2018).

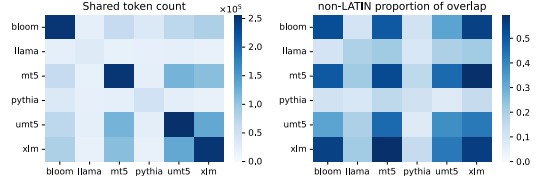

Figure 2: There is substantial overlap in vocabulary between models (mean 60.4%), but much of the overlap is Unicode LATIN. mT5 and XLM-R have the largest non-Latin intersection (59.4%). While large, BLOOM vocabularies do not include Slavic languages, Japanese, or Korean.

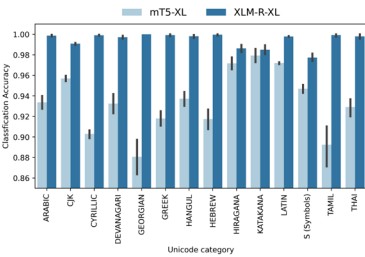

Figure 3: Token embeddings can predict Unicode categories. Both models have high accuracy, but XLM-R-XL is significantly higher than mT5-XL across the majority of categories.

ies, they are often comparable. Current vocabulary sizes of monolingual models are often ≈ 32,000 (BERT, T5, LLaMa), 50,000 (GPT2, GPT-J, Pythia), or 65,000 (falcon). Highly multilingual models have larger sizes around 250,000 (mT5, umT5, XLM-R, BLOOM).

To make strings comparable across tokenzation methods, we replace the space character in BPE tokenizations with Unicode LOWER ONE EIGHTH BLOCK, which looks like a long underscore, as it is more readable and compatible with T5-style SentencePiece vocabularies.

Figure 2 shows the total overlap in vocabularies between six LLMs. We select mT5 and XLM-R as the most multilingual comparison due to their large vocabulary size, significant overlap, and large overlap in non-LATIN tokens. Of the models with a large vocabulary, BLOOM focuses more on African and South Asian languages and thus has a much smaller token overlap in CYRILLIC, HANGUL, HIRAGANA, and KATAKANA.

## 4   Results

**Models encode languages differently.**   We make comparisons between the embeddings of XL-scale

models from mT5 and XLM-R families. In addition to -XL models being potentially more expressive, XLM-R-XL and mT5-XL are also more comparable in parameter size (3.5B and 3.7B, respectively). Figure 1 shows UMAP (McInnes et al., 2018) projections of the embeddings from XLM-R-XL and mT5-XL for each token in the shared vocabulary, colored by Unicode category. We find that XLM-R's representations encode languages — tokens of different categories form isolated clusters. Clusters of Unicode categories are also noticeable in mT5 but they are more overlapping. The exception is Symbols (S) and Numbers (N): they are scattered in XLM-R-XL, but clustered in mT5-XL.

To further show how embeddings encode languages, we use logistic regression to predict Unicode category from embeddings. For each category, we construct a balanced dataset and run 10-fold cross-validation. Embeddings from XLM-R-XL and mT5-XL both encode a high level of language information. Tokens with different Unicode category could be linearly separated with an average of 99.24% and 93.32% accuracy, respectively. Figure 3 shows accuracy for selected categories. XLM-R-XL has significantly higher (near perfect) accuracy across the majority of categories than mT5-XL.

**Embedding neighborhoods encode semantics across languages.**   We next zoom in to study the neighbors of individual tokens. Neighbors from mT5-XL are often direct translations, whereas XLM-R-XL finds tokens in the same language that are semantically or linguistically similar. Figure 5 shows that the 50 nearest neighbors for mT5-XL tokens have an average of 7.61 different Unicode categories, whereas XLM-R-XL has 1.64. Figure 4 shows examples of 20 nearest neighbors of selected tokens.

The neighboring categories of tokens in mT5 vary by the category of a token, as shown in Figure 6. We find interesting comparisons between two Japanese writing systems, KATAKANA and HIRAGANA. KATAKANA, a writing system often used for words of foreign origin, has more diverse neighbors (most often LATIN). HIRAGANA, used more for native Japanese words, has more neighbors in HIRAGANA and CJK (Kanji). We do not find evidence to suggest that tokens appearing more often in pre-training corpora have more diverse neighbors (Figure 9 in the Appendix).

| token | XLM-R-XL neighbors | mT5-XL neighbors |
|---|---|---|
| Comment | Comments, Review, Blog, Update, Text, Group, Info, Support, Photo, Share, Work, Information, Article, Link, Video, Chat, Search, News | Comments, komment *("comment", is)*, Kommentar *("comment", sv)*, omentário, Commentaire *("Comment", fr)*, Коментар *("Comment", bg)*, Kommentare *("Comments", de)*, Komentar *("Comment", ms)*, Koment *("Commentary", sq)*, kommentarer, comentário *("comment", es)*, coment |
| Nike | Adidas, Sony, Samsung, Rose, Mini, BMW, Apple, NBA, Toyota, Blue, Mike, Honda, Volkswagen, Green, Max, Puma, Alex, 55, Ferrari | Adidas, ナイキ *("Nike", ja)*, Asics, Reebok, Puma, نايك *("Nike", ar)*, Converse, SICS, Jordan, 나이키 *("Nike", ko)*, sneaker |
| miliki *("possess", ms)* | mempunyai *("have", ms)*, shika *("hold", sw)*, memiliki *("possess", ms)*, hitaji *("need", sw)*, sajili *("register", sw)*, Li, gunakan *("use", ms)*, shiriki *("participate", sw)*, Ka, lihat *("look", ms)*, lakukan *("do", ms)*, bawa *("wing", sw)*, hifadhi *("reserve", sw)*, ishi *("live", sw)*, shauri *("to advice", sw)*, bentuk *("shape", ms)*, kata *("cut", sw)*, gambar *("picture", ms)* | punya *("have", ms)*, milik *("posession", ms)*, 拥有 *("own", zh)*, lakukan *("do", ms)*, berikan *("give", ms)*, tiene *("has", es)*, makai, sahip *("owner", tr)*, possu, ပိုင်ဆိုင်, ilki, irklich, เป็นเจ้าของ *("own", th)*, enggunakan, មាន *("have", km)*, 具備 *("possess", zh)*, ပိုင်ဆိုင် *("own", my)*, มี *("have", th)*, ediakan, มีความ |
| när *("when", sv)* | when, når *("when", da)*, då *("then", sv)*, eftersom *("since", sv)*, två *("two", sv)*, nær *("closer", is)*, där *("where", sv)*, för *("for", sv)*, första *("first", sv)*, från *("from", sv)*, här *("here", sv)*, också *("also", sv)*, nästan *("almost", sv)*, även *("even", sv)*, är *("are", sv)*, innan *("before", sv)*, stora *("large", sv)*, människor *("people", sv)* | når *("when", da)*, nær *("closer", is)*, when, lähe *("go", et)*, cuando *("when", es)*, quando *("when", pt)*, ווען *("when", he)*, för *("for", sv)*, innan *("before", sv)*, near, att *("to", sv)*, Wenn *("if", de)*, hur *("how", sv)*, quand *("when", fr)*, quan |
| アメリカ *("America", ja)* | 米国 *("USA", ja)*, イギリス *("England", ja)*, 韓国 *("Korea", ja)*, 미국 *("USA", ko)*, フランス *("France", ja)*, ドイツ *("Germany", ja)*, イタリア *("Italy", ja)*, アジア *("Asia", ja)*, 日本の *("Japanese", ja)*, ロシア *("Russia", ja)*, ヨーロッパ *("Europe", ja)*, 東京 *("Tokyo", ja)*, 海外 *("abroad", ja)*, 美国 *("USA", zh)*, 英国 *("England", zh)*, أمريكا *("America", fa)*, امریکا *("America", fa)*, 일본 *("Japan", ko)*, 美國 *("USA", zh)*, 現代 *("modern", zh)* | アメリカの *("American", ja)*, 미국 *("USA", ko)*, Amerika *("America", ms)*, 米国 *("USA", ja)*, 美国 *("USA", zh)*, イギリス *("England", ja)*, Америка *("America", mk)*, ಅಮೇರಿಕ *("America", ka)*, 美國 *("USA", zh)*, フランス *("France", ja)*, அமெரிக்க *("American", ta)*, America, അമേരിക്ക *("America", ml)*, amerikansk *("American", da)*, amerikanische *("American", de)*, meerika, อเมริกา *("America", th)* |

Figure 4: The top 20 neighbors from mT5-XL are often direct translations; whereas those from XLM-R-XL are often semantically or linguistically similar words in the same language. The last row is an example of KATAKANA, the Japanese writing system used for words of foreign origin. ISO 639-1 language abbreviations are included in the Appendix. Some words exist in several languages, we record only one. Closely related languages are often neighbors, e.g. Swedish (sv) and Danish (da). Malay (ms) words are translated from root terms.

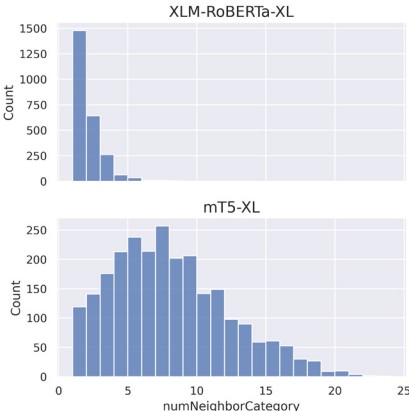

Figure 5: The 50 nearest neighbors for mT5-XL tokens have an average of 7.61 Unicode categories, and XLM-R-XL has 1.64. The figure is made by randomly selected 1% of tokens.

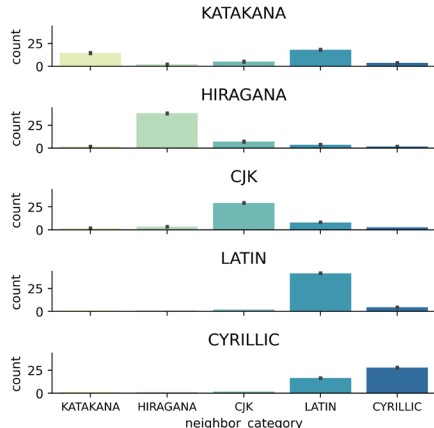

Figure 6: In the 50 nearest neighbors of mT5-XL tokens, HIRAGANA and LATIN find the majority of neighbors in their respective writing systems; whereas KATAKANA and CYRILLIC tokens have more diverse neighbors.

**Embedding geometries are similar across parameter scales.** Finally, we consider whether initial token embeddings as a whole are similar across model families and parameter scales. We use two metrics to quantify similarity in geometric structures of embedding space.

The first metric measures the local similarity between vectors in two matrices. We first find the set of tokens shared between model families and create subset embedding matrices that only

include those tokens. For each token in the shared vocabulary, we then find the 100 nearest neighbors within each matrix using cosine distance. Finally, we calculate the overlap between these neighbor sets from each pair of models. For example, on average, 60 of the 100 nearest neighbors of a token in XLM-R-XL will also be nearest neighbors of the same token in XLM-R-XXL. Figure 7 compares the average number of overlapping terms out of 100

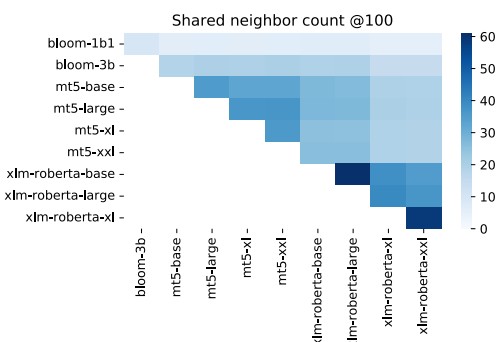

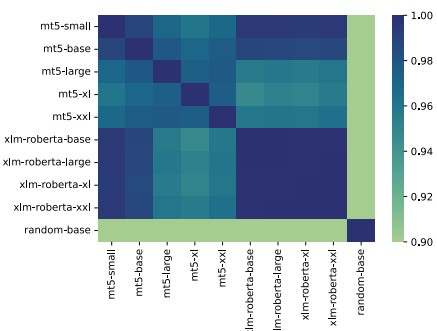

Figure 7: There is a high level of overlap between the nearest neighbors of tokens, derived from embeddings. XLM-R models are the most similar, with up to 60 out of 100 shared neighbors. mT5 is less consistent, and BLOOM is significantly different.

Figure 8: Embeddings for overlapping tokens have similar geometries, as measured by canonical angles. XLM-R models are extremely similar to each other and mT5 small and base. All models are far from random (0.14–0.27).

nearest neighbors for pairs of models, including BLOOM for comparison. Results are averaged over 45,000 shared tokens between mT5, XLM-R, and BLOOM. We find that XLM-R models have the largest similarity. mT5 are more varied, but still find 20–30 shared tokens on average. BLOOM is the most different: the 1.1B model shares only 5–10 tokens with other models, the 3B 15–20.

The second metric uses canonical angles, a linear algebraic measure of rotational alignment. While it is extremely unlikely that input embedding matrices will be comparable at the level of individual columns or entries, the matrices of two models may be identical up to rotation or permutation. We can measure the degree to which two matrices are rotations through the method of canonical angles. Given matrices $A$ and $B$ both with $n$ rows, we calculate $Q_A R_A = A$ and $Q_B R_B = B$. Then we form the SVD $U\Sigma V^T = Q_A^T Q_B$. The singular values $\Sigma$ are the cosines of the angles of rotation. The single largest value is the cosine of the smallest angle, so it forms an upper bound on our ability to rotate one matrix to match the other. Figure 8 shows the first singular values for pairs of models restricted to rows for their shared vocabularies, including a random "embedding" matrix with size 512. We find that XLM-R models have a high rotational similarity to each other (0.99 out of 1.0), while mT5 models are more differentiated (0.95–0.97) but still highly similar. All models are significantly more similar to each other than to a random matrix (0.15–0.27).

## 5 Conclusion

While input embeddings and sub-word tokenized vocabularies of LLMs may appear inscrutable, we find that they are in fact interpretable and meaningful. We observe significant patterns that differ by model families, including an emergent ability of mT5 to discover a shared semantic space that spans languages — accidentally achieving a goal that defied a decade of research on cross-lingual word embeddings.

Future directions include explaining factors that cause the different token embedding patterns we observe in XLM-R and mT5. This could include investigations in model architectures, pre-training procedures, and data-centric strategies such as the curation of pre-training corpora. Finding an efficient path to a more equitable language model performance will be valuable for enhancing language technology for low-resource languages. An interesting next study could explore the utility of combining low-resource languages with closely related higher-resource languages in pre-training corpora. Another future direction is to explore the potential applications of cross-lingual representations embedded in LLMs — What does it say about downstream applications? Could it be used to guide practitioners to select models that are more appropriate for their intended uses?

## 6 Limitations

This work is *descriptive* rather than explanatory. We observe that there are patterns in the geometric structure of input embedding matrices in families of LLMs, but we are unable to identify *why* these

patterns emerge and differ. There are many differences in model architectures, training methods, and pre-training corpora between LLM families. It is out of the scope of this work to determine what factors are causal. As we have limited ability to carry out pre-training experiments, we chose to focus on descriptive observations of existing LLMs.

## Acknowledgements

We would like to thank the anonymous reviewers for their valuable comments. We'd also like to thank Inle Bush, Hyunju Kim, Omary Mzava, Daniel Mwesigwa, Cheng Perng Phoo, Top Piriyakulkij, and Ahkln Wong for their assistance in vocabulary translation.

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

## A ISO 639-1 language codes

The language codes for Figure 4: ar: Arabic, bg: Bulgarian, da: Danish, de: German, es: Spanish, et: Estonian, fa: Persian, fr: French, he: Hebrew, hu: Hungarian, is: Icelandic, ja: Japanese, km: Khmer, ko: Korean, mk: Macedonian, ml: Malayalam, ms: Indonesian/Malay, my: Burmese, pt: Portuguese, ru: Russian, sw: Swahili, sv: Swedish, sq: Albanian, tr: Turkish, th: Thai, zh: Chinese

## B Frequency of tokens does not correlate with diversity of neighbors.

It could be that words with fewer occurrences in the training corpus would have more or less diversity in Unicode categories in their neighbor sets. We calculated an estimate of the frequency of mT5 SP tokens based on a sample from the mC4 dataset. We then took a stratified sample of tokens from the mT5 vocabulary from 10 frequency bands and calculated the mean number of distinct Unicode categories for their neighbors, see Figure 9. We

find no correlation between the frequency of terms and the diversity of their nearest neighbor sets.

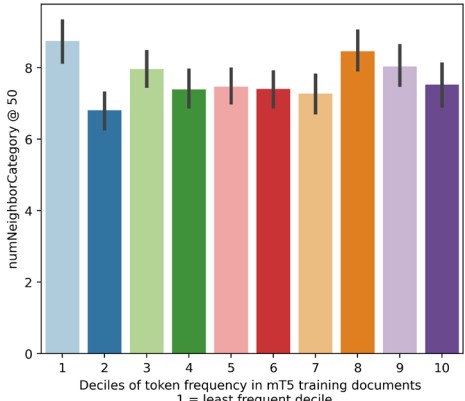

Figure 9: There is no correlation between how often a token is found in pre-training corpora and how diverse a token's 50 nearest neighbor set is. The neighbor set is calculated with mT5-XL embeddings.