# OpenReview forum: "Hyperpolyglot LLMs: Cross-Lingual Interpretability in Token Embeddings"
_EMNLP/2023/Conference — EMNLP 2023 Main_

### Official Review · Reviewer_E6Zp · 2023-08-05

**Soundness:** 3

**Excitement:**

3: Ambivalent: It has merits (e.g., it reports state-of-the-art results, the idea is nice), but there are key weaknesses (e.g., it describes incremental work), and it can significantly benefit from another round of revision. However, I won't object to accepting it if my co-reviewers champion it.

**Paper Topic And Main Contributions:**

This paper studies the cross-lingual interpretability of token embeddings by analyzing the distribution of vector space. The experiments demonstrate that: 1) the distribution of Unicode categories differs among model families; 2) embedding geometries are similar across scales and model families.

**Reasons To Accept:**

1. This paper is well-organized and easy to follow.
2. The idea of studying cross-lingual interpretability is interesting.
2. The analysis in this paper is appropriate and adequate.

**Reasons To Reject:**

The findings of this paper are somewhat weak. Firstly, the vector distribution of different categories is not consistent among model families. Moreover, the findings that "embedding neighborhoods encode semantics" and "embedding geometries are similar across scales and model families" are not new to me.

**Reproducibility:**

3: Could reproduce the results with some difficulty. The settings of parameters are underspecified or subjectively determined; the training/evaluation data are not widely available.

**Reviewer Confidence:**

3: Pretty sure, but there's a chance I missed something. Although I have a good feel for this area in general, I did not carefully check the paper's details, e.g., the math, experimental design, or novelty.

---

> ### Author Rebuttal · Authors · 2023-08-28
>
> Thank you for your review. We are delighted that you found our work in cross-lingual interpretability of embedding layers interesting and easy to follow!
>
> Regarding the significance of the work: the result we are most excited about is that some (but not all) models encode semantics *across languages* despite having no supervision to do so and no explicit parallel cross-lingual training data. We can clarify this in a final version. As in many cases, we will also try to emphasize that there are several possible hypotheses for how embedding vectors are organized, any one of which might have seemed "obvious" in retrospect.
>
> Regarding our findings being not new, we have had difficulty finding any comparable work and would be happy to include any references that can be suggested.
>
> With respect to the vector distribution of different categories not being consistent among model families – this is exactly what we find interesting and important in our work! In both our introduction and Figure 1, we emphasize that embeddings of XLM-R encode language first and then semantics, while that of mT5 encode semantics across languages. We did not know this property in advance and cannot currently explain it. We believe this discovery will be useful in guiding future development of language models, especially for low-resource languages.

---

### Official Review · Reviewer_KmTx · 2023-08-10

**Soundness:** 3

**Excitement:**

4: Strong: This paper deepens the understanding of some phenomenon or lowers the barriers to an existing research direction.

**Paper Topic And Main Contributions:**

The authors investigate the questions of how multilinguality is encoded in the embedding layers of different multi-lingual LMs.
They compare XLM-R and mT5 in more detail (but also other models) and find that XLM-R encodes languages by writing systems, while mT5 rather aggregates similar words in the same areas, no matter their language.
This is interesting, since these models were not trained on parallel data.
They find that the XLM family
- encodes languages into more isolated clusters (fig 1)
- achieves a higher accuracy on unicode prediction (fig 2)
- shows higher language unification among tokens' neighbors (fig 3)
- neighbors tends to be synonym than translation (fig 4)

Comparing model families, the XLM family encodes the embedding the most alike.
Further, the XLM family seens to have the highest overlap between the nearest neighbors of token embeddings followed by the mT5 family. Finally the BLOOM family has the most variants.
The XLM family embeddings are more alike than those of mT5 family (appendix fig 8)


**Questions For The Authors:**

- Not necessarily a question, but it might also be interesting to compare the models wrt their "type", i.e., decoder-only vs. encoder-only models. For this, you might add multi-lingual BERT, for example.


**Reasons To Accept:**

- The paper provides interesting insights and sheds some light on the differences between the encodings of different models.
- Although the results are not "super surprising", they might help to choose models depending on their characteristics. The findings of the paper might be an interesting add-on to, e.g., the model cards on huggingface.


**Reasons To Reject:**

- The paper is rather descriptive, as the authors say themselves, in that they only *show* the described characteristics, and do not explain them.
- The authors should add some ways how to use their insights: What can we learn from knowing the difference of encoding styles (mT5-XL, XLM-R-XL). Can we learn when to select which model or how to leverage them better differently? Similarly, what can we learn from the embedding similarity among families?

**Reproducibility:**

4: Could mostly reproduce the results, but there may be some variation because of sample variance or minor variations in their interpretation of the protocol or method.

**Reviewer Confidence:**

3: Pretty sure, but there's a chance I missed something. Although I have a good feel for this area in general, I did not carefully check the paper's details, e.g., the math, experimental design, or novelty.

**Typos Grammar Style And Presentation Improvements:**

- Some of the plots would be nice to be in the main part of the paper, e.g., Figure 6, maybe you can shorten some of the text.

---

> ### Author Rebuttal · Authors · 2023-08-28
>
> Thank you so much for your input! We are glad that you find our insights interesting and particularly appreciate that you checked our figures in the appendix – we would certainly love to move some of the content currently in the appendix to the main paper in a final version.
>
> Regarding how to use our insights: While our goal in this short paper is interpreting encodings of multilingual LLMs, we can further clarify directions for future work, taking inspiration from the questions raised here. Specifically, future directions could include explaining what contributes to the different geometry of embeddings we observe in XLM-R and mT5. This could include investigations in model structure (encoder/decoder) as well as data-centric strategies such as curation of pretraining data. This would be especially valuable for enhancing language technology for low-resource languages where the current data-hungry methods do not work well. Another future direction will be how to use our knowledge about the embedding geometry of multilingual LLMs – For example, what does it say about downstream applications? Could it be used to guide practitioners to select models that are more appropriate for their intended uses?

---

### Official Review · Reviewer_Eu6p · 2023-08-11

**Soundness:** 4

**Excitement:**

4: Strong: This paper deepens the understanding of some phenomenon or lowers the barriers to an existing research direction.

**Paper Topic And Main Contributions:**

The paper investigates the token emebdding layer of multilingual LMs. It automatically investigates the distribution of writing systems. It also trains a language classifier for scripts with 1-1 language-script mappings, and looks at the nearest neighbours both automatically and with manual investigation. Interesting differences are found between the embedding layers of different popular multilingual LMs.

**Reasons To Accept:**

I think this is a very well-written and dense short paper. It answers all the questions about its specific topic that could be answered both in feasible time and in 3 pages, and includes both experiments at scale and closer detailed looks. While I'm not familiar enough with the literature to be sure that there is no prior work on this, on the assumption that there isn't, this seems to me a specific and confused contribution in the best sense. The authors acknowledge that it would be much more interesting to create experiments to determine why the embedding spaces are so different and how that might affect the model performance, but I agree that it would be out of the scope of this paper.

**Reasons To Reject:**

I can't immediately think of any (beyond possibly the limited scope, which would seem unfair for a very dense short paper), but I'm not familiar enough with the literature to exclude the possibility that they exist.

**Reproducibility:**

3: Could reproduce the results with some difficulty. The settings of parameters are underspecified or subjectively determined; the training/evaluation data are not widely available.

**Reviewer Confidence:**

2: Willing to defend my evaluation, but it is fairly likely that I missed some details, didn't understand some central points, or can't be sure about the novelty of the work.

**Typos Grammar Style And Presentation Improvements:**

Is something wrong with the references? Sometimes they just say "In ACL". I recommend to use the official ACL anthology bibkeys wherever possible.

---

> ### Author Rebuttal · Authors · 2023-08-28
>
> Thank you for the reviews! We are particularly grateful for the recognition of what can be accomplished in the scope of a short paper. We are not aware of prior work exploring the interpretability of multilingual LLMs’ initial embedding layer. As we put in the related works section, prior works on building cross-lingual word embeddings relied on supervision and are sensitive to language pairs; we believe this is what makes our findings exciting: language models construct embeddings that encode cross-lingual semantic relationships without supervision and with multiple languages concurrently.

---

### Meta-Review · Area_Chair_rBa7 · 2023-09-20

**Recommendation:** 4

**Metareview:**

This work examines the input (non-contextual) embedding structure of two families of multilingual LMs, finding that one (XLM-RoBERTa family) learns to separate languages (as one might guess) while the mT5 family unsupervisedly learns cross-lingual structure in the embedding space, as nearest neighbors tend to be related words in other languages. Reviewers praised it for being well-written and interesting, but had some concerns as to the lack of concrete takeaways, a somewhat small scope, and the importance of the results relative to existing knowledge of cross-linguality in language models.

Overall, I think that it is a reasonable contribution for a short paper, but I would challenge the authors to engage with the question of what the importance of these results are for the broader goal of understanding the models–do they suggest an interesting next study? Are they related to an overall improved cross-lingual transfer behavior in mT5?

---

### Decision · Program_Chairs · 2023-10-07

**Decision:**

Accept-Main

**Comment:**

This work examines the input (non-contextual) embedding structure of two families of multilingual LMs, finding that one (XLM-RoBERTa family) learns to separate languages (as one might guess) while the mT5 family unsupervisedly learns cross-lingual structure in the embedding space, as nearest neighbors tend to be related words in other languages. Reviewers praised it for being well-written and interesting, but had some concerns as to the lack of concrete takeaways, a somewhat small scope, and the importance of the results relative to existing knowledge of cross-linguality in language models.

Overall, I think that it is a reasonable contribution for a short paper, but I would challenge the authors to engage with the question of what the importance of these results are for the broader goal of understanding the models–do they suggest an interesting next study? Are they related to an overall improved cross-lingual transfer behavior in mT5?